# Key Proteomics Tools for Fundamental and Applied Microalgal Research

**DOI:** 10.3390/proteomes12020013

**Published:** 2024-04-04

**Authors:** Maxence Plouviez, Eric Dubreucq

**Affiliations:** 1School of Agriculture and Environment, Massey University, Palmerston North 4410, New Zealand; 2The Cawthron Institute, Nelson 7010, New Zealand; 3Agropolymer Engineering and Emerging Technologies, L’Institut Agro Montpellier, 34060 Montpellier, France; eric.dubreucq@supagro.fr

**Keywords:** microalgae, biochemical pathways, biotechnology, proteomics, high-throughput, protein modelling

## Abstract

Microscopic, photosynthetic prokaryotes and eukaryotes, collectively referred to as microalgae, are widely studied to improve our understanding of key metabolic pathways (e.g., photosynthesis) and for the development of biotechnological applications. Omics technologies, which are now common tools in biological research, have been shown to be critical in microalgal research. In the past decade, significant technological advancements have allowed omics technologies to become more affordable and efficient, with huge datasets being generated. In particular, where studies focused on a single or few proteins decades ago, it is now possible to study the whole proteome of a microalgae. The development of mass spectrometry-based methods has provided this leap forward with the high-throughput identification and quantification of proteins. This review specifically provides an overview of the use of proteomics in fundamental (e.g., photosynthesis) and applied (e.g., lipid production for biofuel) microalgal research, and presents future research directions in this field.

## 1. Introduction

Photosynthetic microorganisms can be found in all aquatic ecosystems, in which, as primary producers, they have important ecological functions, either positive (e.g., biogeochemical cycle [1,2,3]), or negative (e.g., when they bloom at the wrong time and place [4,5]). Although they encompass both prokaryotes (cyanobacteria) and eukaryotes (“true” microalgae), they will be collectively referred to as microalgae in the present review [6]. Microalgae such as the model *Chlamydomonas reinhardtii* are widely studied to improve our understanding of key plant functions such as chloroplast-based photosynthesis and nutrient assimilation, and the structure and function of the eukaryotic flagella [7,8,9,10]. In addition, due to their physiology and biochemistry, significant research is focusing on the development of microalgae-based environmental biotechnologies for food and high-value molecules production, biofuels generation, wastewater treatment and nutrient recovery [11,12,13,14].

In recent years, the so-called “omics” technologies have been shown to be valuable tools in microalgal research [15,16,17,18]. Omics, which includes genomics, transcriptomics, proteomics and metabolomics, broadly refers to the comprehensive analyses of classes of biological molecules, i.e., DNA, RNA, proteins and metabolites, and their interactions (interactomics, Figure 1). Omics allow us to determine and study the whole makeup of a cell/biological system at a given time and, therefore, to correlate molecular signatures with phenotypes [17,19,20,21,22,23]. With a wide range of applications, these technologies have considerably accelerated the rate of discoveries and provided a leap forward in fundamental and applied research. For example, the sequencing and study of the *C. reinhardtii* genome in 2007 unraveled the evolution of the eukaryotic flagellum and plastid [7]. In the past decade, technological developments in hardware and software have allowed omics technologies to become more affordable and more accessible [22,24,25]. Consequently, it is now possible to study the whole genome, transcriptome, proteome and metabolome of an organism in a matter of weeks (assuming the proper steps are considered and followed). As clear evidence, whereas until 2008 the genome of only three microalgae had been sequenced [16], many omics datasets have now been deposited in public repositories.

While genomics and transcriptomics deliver significant information about genes and their expressions, these approaches do not provide an indication of protein levels, protein turnover and post-translational modifications. Consequently, proteomics, i.e., the study of proteins (from their structure to their interactions with other molecules in a cell), corroborates and builds on genomics and transcriptomics and it is, therefore, a comprehensive approach to characterize a biological system. As proteins are effectors of biological functions, the levels (and forms) of proteins in a cell indeed represent comprehensive information about cellular function, defining the phenotype of a cell in response to genetic or environmental changes [13,19,25].

This review aims to give an overview of proteomics technologies and their applications for microalgal research and focuses on the recent improvements in the use of proteomics in microalgal research. It also presents the future of proteomics in the post-omics era, i.e., the development and use of artificial intelligence (AI)-based technologies. Recent advancements in machine learning, particularly deep learning, have enabled researchers to accurately predict protein structures and functions from their amino acid sequences, improving the quality and reliability of analytical workflows in mass spectrometry-based proteomics [26]. This approach is becoming central to biomarker discovery from proteomics data, and is beginning to outperform existing assays [27]. Furthermore, AI algorithms have been instrumental in analyzing large-scale proteomic datasets [27,28,29,30], ultimately enabling the identification of novel protein interactions and pathways that are critical for microalgal adaptation to environmental stresses [16,31]. It is noteworthy that the potential of microalgal biotechnology and that of proteomics technologies, along with their respective advantages and disadvantages, have been reviewed elsewhere [11,16,19,21,32,33]. Therefore, these topics are not the primary focus of this review.

## 2. Proteomics

According to the central dogma of molecular biology, the genetic information encoded in the nucleotide sequence of DNA is transcribed into messenger RNA (mRNA). This mRNA is then translated into a polypeptide chain of amino acids. Subsequently, the polypeptide undergoes folding, often facilitated by chaperone proteins, as well as post-translational modifications. These processes, along with potential interactions with other proteins or molecules, lead to the formation of a functional protein [19,20,21,25,33]. The proteome is defined as the whole set of proteins of a biological system, and proteomics refers to methodologies to identify, characterize, structurally study and/or quantify proteins (further details in Box 1, and see examples of methods used in microalgal research in the next section). 

Methodological advancements, including the development of mass spectrometry techniques for precise mass and chemical structure analysis, have revolutionized protein research. In conjunction with growing peptide and protein sequence databases as well as advanced bioinformatics tools for handling and analyzing large datasets, they have enabled rapid and accurate large-scale analyses of proteins [19,21,25,31], a significant shift from the study of single or a few proteins that characterized the field decades ago. 

Proteins are complex molecules that serve structural roles, as well as drive and regulate metabolic reactions within cells. Studying the proteome is therefore ideal to characterize a biological system at a particular time or phase [19,21,33]. In addition, as proteins perform a function and are encoded in the genome, proteomics is considered one of the most significant methodologies to characterize and understand gene functions [19]. However, proteomics can be challenging, as proteomes are significantly more complex than genomes because of the vast range of proteins found at different abundances and in different forms (i.e., iso/proteoforms) according to post-translational modifications [19,21,23,34]. This diversity also implies a tremendous number of possible interactions between proteins and between proteins and other molecules. These interactions are the subject of the more recent field of interactomics, which attempts to assess as comprehensively as possible this network of interactions [2,20,35,36] (Figure 1). 

The variations in protein turnover time and the influence of post-transcriptional and post-translational regulations can also contribute to the complexity of proteomes. This can explain discrepancies between the transcriptome (i.e., the abundance of transcripts) and the proteome (i.e., the abundance of proteins), as reported for the microalgae *Thalassiosira pseudonana* [37] or *Chlamydomonas reinhardtii* [38]. Indeed, moderate correlations of ~0.4 have been widely reported between the abundance of proteins and their corresponding mRNAs in both prokaryotes and eukaryotes [34,39]. Conversely, Plouviez et al. [40] recently showed, by temporal analysis of both datasets, that the transcriptome and the proteome related to phosphorus metabolism are synchronized in *Chlamydomonas*. This provided insight into the turnover and regulation of Vacuolar Transport Chaperone proteins, key proteins involved in polyphosphate synthesis in microalgae [40,41,42].

The knowledge provided by proteomic research adds greatly to the genetic information accumulated from genomics and transcriptomic studies, as the field of proteomics is broad and encompasses many different interrelated areas of study, including expression proteomics, functional proteomics and structural proteomics (Box 1).

Box 1Main field of proteomics.  *Expression proteomics*: Refers to the measurement and comparison of protein abundance in the entire proteome (or subproteome) between samples that differ by some parameters. For example, the proteome of cells cultivated under different conditions can be compared, as was done by Plouviez et al. [40] with *C. reinhardtii* cultures grown under phosphorus-depleted and repleted conditions. The proteome of two different phenotypes can also be compared, as was done by Sithtisarn et al. [43] with a salt-tolerant mutant and “wildtype” *Chlamydomonas* strains. Expression proteomics complements and builds on proteome identification and mining by identifying proteins that are normally expressed in a cell or organelle at a specific time or phase [44], as in the case of the eyespot of *C. reinhardtii* [45].  *Functional proteomics*: A broad term for many specific, directed proteomics approaches to identify the biological functions of specific proteins or classes of proteins. Due to their key role in functional proteomics, significant research is focused on post-translational modifications (PTMs) of proteins, such as oxidation, phosphorylation or acetylation [20,33,46,47,48,49]. Indeed, these modifications regulate protein activity, stability, localization and interactions within the cell [20,33,46]. Specific proteomic studies focused on PTMs in *C. reinhardtii* have revealed the complexity of the redox protein regulation network in this microalga and the importance of PTMs in protein regulation and signal transduction during stress responses [46,47,50,51,52,53].  *Structural proteomics*: This area of proteomics focuses on characterizing the structure of proteins or protein complexes. The information gathered helps determine how a protein works and how it is regulated, as was done by Gurrieri et al. [54] for *C. reinhardtii* phosphoribulokinase or by Shen et al. [55] for the PSII-LHCII supercomplex involved in light-induced oxidation of water during photosynthesis. Ultimately, structural studies can provide critical information on protein function. Due to ongoing technological advancements, such as mass spectrometry (MS) analyses, protein–protein interaction network analyses, and proteome-wide imaging of protein localization [31,56,57], it is now possible to identify and locate all proteins within a protein complex or organelle [44,57,58,59]. Additionally, these technologies enable the characterization of protein–protein interactions (how, which, when and where proteins interact). This field, known as spatial proteomics, allows us to study the overall architecture of organelles and elucidates the roles that proteins play in cellular pathways and organellar dynamics [56,60].

## 3. Overview of Proteomics Techniques Applied in Microalgae Research

Proteomics can be hypothesis-driven or hypothesis-generating depending on if a proteomic experiment targets certain known proteins of interest (direct approach), or instead measures all the proteins present in a sample (defined as the indirect or shotgun approach) [61]. 

Table 1 presents the main methodologies used to study individuals, groups or the whole set of proteins in a sample. The choice of methods is highly dependent on the aim of the study, and often multiple methods are used in parallel, such as in the study reported by Stauber et al. [62]], which used 2D gel electrophoresis to separate *C. reinhardtii* light-harvesting proteins and identify these proteins using mass spectrometry. Data outputs can either be qualitative (i.e., is a protein present) or quantitative (i.e., is a protein found in higher abundance in one particular sample). Notably, there has been significant development in, and interest toward, quantitative high-throughput methodologies in recent years [18,20,21]. As a result, conventional methods are nowadays primarily used for specific applications (i.e., protein separation) or for data validation (Figure 2).

### 3.1. Quantitative High-Throughput Proteomics: Mass Spectrometry

Mass spectrometry is now commonly used to identify and quantitatively measure protein levels in microalgae samples. MS quantitative proteomics proved useful in fundamental and applied microalgae research (Section 5) to compare the physiological state of different microalgae strains or microalgae grown under different conditions. While top-down (i.e., intact proteins, [76]) and bottom-up (i.e., digested proteins, [77]) analyses are alternative strategies for protein identification and characterization by mass spectrometry, bottom-up approaches have mostly been used for microalgal proteomics.

Label-based and label-free strategies can be used for protein quantification. Label-based techniques (such as ICAT, iTRAQ, SILAC, QconCAT, TMT; Table 1) incorporate stable isotope labels into peptides, creating an anticipated change in mass that can be measured. Conversely, label-free proteomics quantify peptide abundance based on signal intensity or spectral counting of peptides [78]. Wang et al. [70], previously compared the use of a label-based method (iTRAQ) and label-free approach on *C. reinhardtii*. The authors found a good overlap of proteins identified with both methods, suggesting both provide high-quality quantitative and qualitative data. While quantification accuracy and reproducibility were better when using iTRAQ than the label-free method, more proteins were identified and quantified with the label-free method. Therefore, the authors suggested that extra steps (e.g., pre-fractionation) might have been needed for a higher coverage when using the iTRAQ method, and they recommended considering the experimental design (i.e., sample numbers, sample complexity, and amount) for selecting a suitable proteomic approach. Notably, label-free methods have been the most popular approaches in microalgal proteomics, potentially because label-free methods are cost-effective and do not require expensive labeling reagents. Independently of the method selected, sample preparation is critical for proteomics analyses, as further described in Section 4.

### 3.2. Protein–Protein Interaction Techniques

Proteins generally drive cellular processes by interacting with other molecules (e.g., metabolites), including other proteins. Protein–protein interactions (PPIs), defined as the physical contacts between two or more proteins, are therefore critical as PPIs can activate or deactivate a specific protein, alter kinetic properties and regulate biological processes. Another key benefit of studying PPIs (or PPI networks) is that it is possible to determine putative functions of uncharacterized proteins. Several methods exist to study PPIs either in vitro or in vivo [79,80]. While X-ray crystallography and NMR are two methods commonly used for in vitro study (see references in Table 1), the yeast two-hybrid assay is widely used in vivo. With this method, the detection of interacting proteins is detected in yeast cells when fused “bait and prey” proteins from a target organism activate reporter genes that enable growth on specific media or a specific reaction leading to color change. This method was successfully applied to characterize the Intraflagellar Transport Complex A Proteins subunit interactions in *C. reinhardtii* [81] or the interaction between the small subunit of Rubisco and the essential pyrenoid component 1, a linker protein required for Rubisco aggregation in algae [82]. The latter improved the understanding of the specific algal CO_2_-concentrating mechanism (CCM). Alternatively, Mackinder et al. [83] used affinity purification–mass spectrometry to measure the interaction of proteins involved in the CCM and the pyrenoid in *C. reinhardtii*. Finally, PPIs (and PPI networks) based on known and predicted interactions can also be studied in silico (further discussed in Section 6). 

Critically, studying PPIs could prove instrumental for strain improvement. For example, using in silico protein modelling and activity-based chemical crosslinking, Blatti et al. [84] evidenced the critical role of the interaction between a fatty acid acyl carrier protein and thioesterase in governing fatty acid metabolism in *C. reinhardtii*, providing an alternative to manipulating fatty acid biosynthesis for algae biofuel. Considering the tremendous potential of microalgae, further research on PPIs of protein involved in metabolic pathways of use for biotechnology would be valuable.

## 4. Key Considerations for Successful Proteomics

As seen in Table 1, a wide range of methods are available to study proteins/proteomes. While each of these methods has its own advantages and limitations [19,33], their success is strongly dependent on protein sample preparation [33,85,86,87]. In addition, especially for high-throughput techniques, huge datasets are being generated and, therefore, require the use of bioinformatics tools and pipelines to acquire and analyze data [24,31,88,89,90]. The following sections, therefore, focus on the importance of sample preparation and bioinformatics in proteomics.

### 4.1. Protein Sample Preparation

Sample preparation generally involves multiple steps (e.g., cell disruption, extraction, fractionation, chemical modifications and purification, e.g., [86,87,91]) that are susceptible to losses and contaminations [77,91,92,93]. Considering the complexity of proteins, their heterogeneity and the cell characteristics, sample preparation can therefore be a time-consuming, limiting step [13,31]. Method optimization is generally required and adds to the labor of sample preparation. Critically, the steps (and options) involved in sample preparation need to be carefully considered to generate quality protein extracts and to prevent failure of downstream analyses. The two sub-sections below focus on protein extraction methods that can be a limiting step in microalgal proteomics and, considering the importance of MS-based techniques in the field, protein sample preparation for high-throughput proteomics.

#### 4.1.1. Protein Extraction

Extraction efficiency varies greatly with the microalgal species considered, due to biochemical and structural differences in membranes [94]. Significant research has been conducted to develop efficient and reproducible sample preparation methods, especially for proteomic analyses of tissues and cells of animals and higher plants [31,86,95,96]. While these protocols have not necessarily been attempted on microalgal biomass, similar cell disruption techniques (i.e., sonication, beading, lysis buffer) and solvents (e.g., trichloroacetic acid, chloroform, methanol) are used for cell lysis and protein solubilization/precipitation, respectively. While proteins can be extracted from wet or dry biomass, lyophilization is generally recommended for sample stability. Anjos et al. [91] recently described a protocol for protein extraction suitable for analyzing the proteome of *Tetraselmis chuii*. However, according to the authors, this protocol effectively generated protein extracts with high yields for larger proteins but was not efficient for proteins with low molecular mass. Importantly, the authors also noted that protein extraction techniques must be specifically selected for each microalgae species. This was also evidenced by Toyoshima et al. [87], who showed that, while sonication in KCl buffer was the most effective method to extract total proteins from *C. reinhardtii*, beads disruption in PTS buffer was better for *Synechocystis*. However, with *Chlamydomonas*, a better detection of individual subunits of the photosynthetic apparatus and other large protein complexes was obtained by the PTS and sonication methods rather than when using the KCl method. Together with the results from Anjos et al. [91], this suggests that, in addition to the species studied, the protocol used for sample preparation must also be selected according to the downstream analyses to be performed. Because of the inherent limitation due to the multiple steps involved for proteomics analyses, as well as the known variability between biological samples, it is highly recommended to conduct preliminary testing and use more than three biological replicates. While there is currently no “standard” protocol for proteomics sample preparation in microalgae, the authors recommend the following literature focusing on microalgal sample preparation: [48,49,85,87,91,93,97].

#### 4.1.2. Sample Preparation for MS-Based Analysis

Bottom-up approaches have mostly been used for microalgal proteomics. For these approaches, extracted proteins are first reduced and alkylated before being digested using proteases to generate peptides that will ultimately be analyzed using MS techniques. While different proteases can be used to broaden proteome coverage, trypsin is the most widely used in proteomics, including in microalgal proteomics [13]. While several proteolytic digestion methods are in development, in-solution digestion is mostly used for microalgae [13]. During in-solution digestion, proteins are digested while remaining in a buffer. Peptide purification/desalting is then applied to remove excess salts and detergents that could interfere with the MS analysis (e.g., [40]). Alternatively to in-solution digestion, in-gel digestion can be performed. In this approach, proteins are separated with gel electrophoresis, and bands or groups of bands of interest are excised and then digested [62]. This allows the analysis of specific groups of proteins, simplify sample complexity and increase the depth of the analysis. However, this process can be time-consuming and prone to error, possibly explaining why in-solution digestion is preferred. Technical progress toward shotgun proteomic analyses, where mixtures of proteins, digested by multiple proteases of different specificities, are simultaneously analyzed by high-resolution MS/MS, has nevertheless modified these practices [98], and the current iterations of this approach have led to the capacity for analyzing and quantitating thousands of proteins per hour from whole-proteome digestions [99,100]. Critically, further investigation of the efficiency and suitability of novel digestion methods and proteases is needed for microalgal proteomics.

### 4.2. Bioinformatics

Bioinformatics encompasses algorithms and software developed to analyze and interpret biological data. For proteomics, bioinformatics can be used for protein–protein search (i.e., classical BLAST), for in silico analyses (e.g., structure modeling and protein-protein interactions as further discussed in Section 6), and to analyze data from, for example, 2D gel electrophoresis [101]. High-throughput proteomics also critically relies on bioinformatics to acquire and analyze the large datasets generated by mass spectrometry [18,23,61,88,90]. 

#### 4.2.1. MS Data Processing

While bioinformatics is critical for both top-down and bottom-up approaches, this review focuses on the bottom-up approach that is commonly used in microalgal proteomics (see Section 3.1). Briefly, a standard bottom-up MS proteomics workflow involves several steps [61,77]:Protein samples are prepared for MS, i.e., solubilized proteins are enzymatically digested (usually using trypsin), chemically modified (alkylation) and purified (de-salting) to obtain short MS-accessible peptides;Peptides are separated using an LC setup that is coupled to a mass spectrometer (LC-MS);Intact peptide masses and the corresponding masses of fragmented peptide ions are measured by mass spectrometry (i.e., tandem LC-MS/MS or MSn setups). As mentioned in Section 3.1, label-based and label-free strategies can be used for protein quantification, and different techniques are available (Table 1). Notably, different data acquisition modes can also be used (e.g., label-free data-dependent acquisition and data-independent acquisition); refer to Schessner et al. [90] for further details;Based on a reference proteome, the resulting peptide and fragment ion spectra are used to identify the peptides present in the sample;Identified peptide sequences are quantified and assembled to measure protein levels by protein inference.Data from MS analyses are susceptible to systematic, dependent or independent biases (e.g., different handling, equipment calibration) on the measured peptide/protein abundances [102]. Therefore, a key step is to normalize the data to take the bias into account, allowing the data to be comparable and downstream analyses reliable [103,104,105]. Advanced analysis pipeline frameworks are therefore needed for data normalization but also for protein inference and data analysis [88,89,103,104,106]. The latter has been extensively reviewed by Schessner et al. [90] in their guide to interpreting and generate visual representation of bottom-up proteomics data.

#### 4.2.2. MS Data Interpretation

Interpreting the large amount of data available from high-throughput datasets such as those for quantitative proteomics can be overwhelming. Based on MS data, thousands of peptides and proteins can be identified and quantified. This allows for the determination of absolute levels within a sample and relative levels across different samples [90]. For the latter, statistical analyses (i.e., Student’s t-test) and fold change calculations (i.e., relative abundance) are used to identify proteins found in higher or lower abundance between treatments and/or phenotypes. For instance, a *p*-value < 0.05 and a fold change of −1.20 to 1.20 could be criteria for significance [40]. Several methodologies can then be followed to gather knowledge from the list of identified proteins [90]. As hundreds of proteins can be statistically found at higher/lower abundance, a common approach is to perform functional enrichment analyses [90,107,108,109]. This bioinformatics method is based on a statistical comparison of a sample dataset against a reference dataset. This allows us to identify classes of proteins (or genes) that are over-represented in a large dataset and may have an association with different phenotypes and/or physiological states. For example, a functional enrichment analysis showed that ribosome synthesis and protein translation are a strong response to the transition from P-depleted to P-repleted conditions in *C. reinhardtii* [40]. 

During functional enrichment analyses, proteins (or genes) are annotated and classified from manually curated libraries on the basis of their function. Gene Ontology (GO) is the most renowned library that provides terms classified under biological processes, molecular functions and cellular components, providing critical information about the location and function of specific proteins [110]. Other libraries also provide alternative classification, such as the KEGG pathway database, allowing the understanding of metabolic pathways influenced by a specific condition [111]. Functional enrichment analyses can be performed using several online software tools (e.g., MapMan [112]; Panther: [40]; see [90,109] for further examples). Notably, because each software tool uses specific algorithms and statistical tests, performing enrichment analyses with several software applications is recommended for data cross-validation [90]. 

When groups of proteins have been identified in response to specific conditions, further studies such as protein–protein interactions network analyses can provide a critical understanding of how proteins work together and which proteins play key functions. For example, a PPI network analysis suggested that proteins involved in protein metabolism, energy supply, and photosynthesis act in synergy to reconstitute cellular homeostasis under salt stress in *Dunaliella salina* [113]. PPI networks can be generated from the STRING software v12 (STRING: functional protein association networks (https://string-db.org/; accessed on 1 April 2024) from its wide database of known and predicted protein–protein interactions.

The continuous effort and use of proteomics profit from the increased number of databases, the availability of data and software, and vice versa. Significant protein information is now collated in databases such as UniProt (https://www.uniprot.org/; accessed on the 1 April 2024), a freely accessible database of protein sequences and functional information. Specific to microalgal research, Predalgo is dedicated to the prediction of protein subcellular localization in green algae [83], and the Algal Protein Annotation Suite (Alga-PrAS: Alga-PrAS (riken.jp)) [114] provides protein databases to enable the interpretation of algal proteome features. The increasing availability of software and databases promises a wide range of discoveries for fundamental research but also for applied microalgal research. Based on these, network-based strategies, while primarily discussed in the context of metabolomics, have also been applied to proteomics [115,116]. These strategies encompass association networks based on quantitative information, mass spectra similarity networks to assist metabolite annotation, and biochemical networks for systematic data interpretation. 

#### 4.2.3. Functional Characterization

Bioinformatics tools have also proved useful to determine the biotechnological potential of proteins identified during high-throughput proteomics. Peptide Ranker, a server for the prediction of bioactive peptides (PeptideRanker (ucd.ie); accessed on 1 April 2024), was used to predict the bioactivity of 500 peptides in the microalga *Tetradesmus obliquus* [117]. Twenty-five peptides with potential antioxidant and angiotensin-converting–enzyme-inhibitory activities were found, allowing the authors to select four of these peptides for further in vitro testing. In another study, Carrasco-Renado et al. [118] identified 488 proteins with potential industrial applications from the proteome of *Nannochloropsis gaditana* from a search of the EMBL’s European Bioinformatics Institute patented proteins database NRPL2 (Non-redundant Patent Sequences < Patent Data Resources < EMBL-EBI, [32,118]).

## 5. The Benefit of Proteomics in Microalgal Research

As in other fields of research, proteomics has been instrumental to many discoveries in algal research [15,16,17]. Most of the proteomics methodologies currently available have been successfully applied to study proteins/proteomes in microalgae, such as *Chlamydomonas* (see references in Table 1). This is not surprising, as *C. reinhardtii* is a model organism not only in phycology but also in plant sciences for key metabolic pathways [8,62,75]. Proteomics approaches are also now widely applied to microalgae species with commercial potential (e.g., *Chlorella vulgaris*, *Nannochloropsis oculata*, *D. salina*), an important step for the development and improvement of microalgal biotechnology. The following sections provide some examples in fundamental and applied microalgae research.

### 5.1. Fundamental Research

*C. reinhardtii* is the first microalga to have been sequenced [7]. Consequently, the availability of extensive resources, such as genetic and proteomic databases, mutant collections and the development of protocols for genetic modifications, make it an organism of choice for fundamental research [8]. Significant research has especially focused on *C. reinhardtii* to study oxygenic eukaryotic photosynthesis [9,38,55,72,119]. Thus, many structural and mechanistic understandings about photosynthesis were discovered in *Chlamydomonas*. As the overall architecture of the photosynthetic core complexes are very similar between *C. reinhardtii* and vascular plants [62,120], findings obtained with *C. reinhardtii* have broad implications in plant biology. 

Proteomics has been widely used to identify and study the proteins involved in photosynthesis in *C. reinhardtii*. For example, the first crystal structure of Rubisco (ribulose-1,5-bisphosphate carboxylase/oxygenase, EC 4.1.1.39: the enzyme binding CO_2_ during photosynthesis) in green algae was generated by X-ray crystallography in *Chlamydomonas* in 2001 [75]. Later, by combining 2D gel electrophoresis and mass spectrometry, Stauber et al. [62] generated a detailed map of *C. reinhardtii* light-harvesting proteins (Lhcas and Lhcbs proteins) and provided a first hint about structural differences in the light-harvesting complexes between green microalgae and vascular plants. The structures of light-harvesting complexes are now available. Particularly, the study of the *C. reinhardtii* PSII-LHCII supercomplex (the light-harvesting complex involved in light-induced oxidation of water during photosynthesis) from cryo-electron microscopy data unraveled protein–protein interactions and key features that explain how microalgae harvest energy efficiently at low light intensities [55]. In the last decade, significant research efforts have focused on characterizing the chloroplast proteome of *Chlamydomonas* [44,57]. A fluorescent protein-tagging and affinity purification–MS pipeline developed in *Chlamydomonas* allowed the identification and the analysis of the interactions of known and new proteins of the pyrenoid (a chloroplast microcompartment found in many algae) with those involved in the CCM, significantly improving our understanding of the algal CCM [83]. In addition, using MS, Zhan et al. [58] characterized 190 proteins, among which 81 were of unknown function, in the pyrenoid proteome of *Chlamydomonas*. Their functional analysis showed that, in addition to its known function to promote photosynthetic CO_2_ fixation by Rubisco, the pyrenoid may be involved in photoacclimation and/or responses to light-induced stress.

Quantitative proteomics has been widely used to improve our understanding of photosynthesis by investigating *C. reinhardtii* wild types or mutants’ responses to changing light [38,72,87,121]. It has also been widely applied to study other metabolisms than photosynthesis in *Chlamydomonas*, such as micronutrient deficiency [122], nutrient use [40,123], salt tolerance [50], etc. Similar approaches were used to study key metabolisms in other microalgae, such as the response of *D. salina* and *Chlorella sorokiniana* to toxic metals [124,125,126] and of *Thalassiosira pseudonana* to P depletion [37]. The latter provided key knowledge on the strategies developed by diatoms to survive P depletion such as swapping phospholipids for sulfolipids to maintain intracellular P levels.

### 5.2. Applied Research

Microalgae farming currently supports a multi-billion USD industry, and significant research is focusing on the development of microalgal biotechnologies for wastewater treatment and the production of high-value molecules and biofuels [11,12,13,14]. Proteomics, and especially quantitative proteomics, could provide insight for cultivation or strain improvement for the above-mentioned biotechnological applications [13,16,18,127,128]. 

Quantitative proteomics has thus been widely applied to microalgae with commercial potential such as *Chlorella* [127,129], *Dunaliella* [113,130,131], *Haematococcus* [132], *Nannochloropsis* [32,133] and *Scenedesmus* [134], to name a few (see Table 1 from [16] and extensive supplementary data from [13]). However, even when considering the tremendous potential of microalgae for biotechnology, an extensive literature search showed that most of the proteomics studies concerned the effects of stresses, such as nitrogen depletion, to trigger lipid production for biofuels [16]. Generally, the majority of differentially abundant proteins are found to have functions in metabolic pathways related to fatty acid and lipid metabolism, carbohydrate metabolism, and photosynthesis [16,17,135]. In brief, during nutrient stress, the cells experience reduced photosynthetic efficiency and divert energy and carbon fluxes toward lipids rather than carbohydrate, protein and chlorophyll biosynthesis. 

In addition to generating knowledge that improves the understanding of the metabolic pathways involved during specific growth conditions relevant to biotechnological applications, quantitative proteomics also allows for the identification or functional characterization of proteins of interest for biotechnology ([13] and see Section 6 below). The next logical step in microalgal biotechnology would, therefore, be to focus on the in silico and in vitro study of proteins with biotechnological potential. Structural and functional information about 31 algal proteomes can now be found in the Algal Protein Annotation Suite (Alga-PrAS) [114]. In addition, several software platforms are now available to determine the biotechnological potential of proteins or characterize specific features (e.g., bioactivity) of proteins with biotechnological potential (Section 4.2).

## 6. Proteomics in the Post-Omics Era

Artificial intelligence (AI) technology, especially based on deep learning, is changing our everyday life. It has been rapidly expanding during the last few years and has already had a strong impact on science [24,27,136]. As mentioned above, bioinformatics play a key part in understanding and unraveling the full potential of proteomics data. Combined with AI, bioinformatics is now on the edge of a huge acceleration and facilitation of data analyses, offering new opportunities and incomparable results in applied research [24,27,29]. The development of AI-based software for MS data analysis is one example. Applications in protein structure prediction is another. AI software such as AlphaFold v2.3.2 or RoseTTAFold v1.0 generate protein structure fast and accurately [137], preventing the need for the laborious, time-consuming and expensive experimental techniques for protein structure determination such as X-ray crystallography [136]. Currently, more than 1 million computed protein structures have been added to the Protein Databank (PDB, https://www.rcsb.org/; accessed on 1 April 2024), the main reference deposit of protein structures, and almost all known proteins have had their structure calculated [138]. Combined with molecular dynamics and ligand docking analyses, in silico protein studies have wide applications in many biological fields such as medicine and drug discovery [136,137,138]. In silico studies are also critical for fundamental research. In the case of microalgae, Cliff et al. [41] used Alphafold 2 and molecular dynamics to generate the first models of VTC4 protein in *C. reinhardtii*, *C. vulgaris*, *Scenedesmus* sp. and *Gonium pectorale*, based on sequences from databases and their own sequencing data (example in Figure 3). Prior to this study, the function of VTC4 as a polyP polymerase was only confirmed in *C. reinhardtii*, without information on its structure. However, as the models showed conservation of the VTC catalytic core and the SPX regulatory domains among the microalgae species and yeast, the study of Cliff et al. [41] confirmed the probable function of VTC4 as a polyP polymerase in the other microalgae. In addition, by showing, through molecular docking calculations, the affinity between inositol phosphates and the SPX domain, this study also suggested a regulation of VTC4 proteins in microalgae that may be similar to that in yeast. These findings have significant implications in microalgal biology and biotechnology, as polyPs have important metabolic functions in microalgae and significant research is focusing on the ability of microalgae to accumulate polyPs for P recovery from waste [4,40,41,42,139].

While protein models inform on the putative functions (e.g., substrates and products) and regulatory features of proteins (and, therefore, on the potential differences in the selectivity and specificity between homologous proteins), this should still be validated experimentally, in vitro. Proteins of interest can be isolated from cell extracts using chromatographic methods (Table 1). However, protein isolation can be hindered by the low concentration of certain proteins in cell extracts and/or by the embedding of some proteins in subcellular structures [37,87,91]. Another possibility is to produce proteins of interest using heterologous expression [140]. In recent years, the advances in synthetic biology have facilitated the process of heterologous expression by providing, through automated gene synthesis and molecular biology pipelines, tools for rapid engineering and production of recombinant proteins. Proteins of interest are overexpressed in a host organism, facilitating the extraction and purification of samples suitable for in vitro studies and/or of products for biotechnological applications (e.g., pharmaceuticals). As for proteomics, heterologous expression consists of several steps that can only partly be automated, i.e., host selection, vector design and cloning, transformation of the host, screening of transformants, cultivation of the transformant, heterologous enzyme production, and finally protein recovery and purification. Several hosts can be used for heterologous expression such as *Komagataella phaffii* (formerly *Pichia pastoris*) and *Escherichia coli* [140,141,142]. The yeast *K. phaffii* is commonly used because of its reliability, the possibility to achieve a high heterologous protein expression level, and the presence of the co-translational and post-translational processing necessary for eukaryotic proteins [140]. Furthermore, the secretory production of recombinant proteins directly into the supernatant of the culture medium is possible with *K. phaffii* [140]. The limited production of endogenous secretory proteins by this yeast is indeed a key attribute for the purification of recombinant protein from supernatants. The microalga *C. reinhardtii* has also been proposed as a promising host to produce recombinant protein [143,144,145]. The successful expression of several proteins with pharmaceutical relevance has been reported from editing of the nuclear and the chloroplastic genome [145]. While promising for biotechnological applications, considerable developments are, however, still needed before it can be used at large scale [143,144].

## 7. Future Prospects

The study of model species like *C. reinhardtii* has shed light on a major challenge faced when dealing with un-sequenced microalgal genomes—the lack of comprehensive databases for protein identification. In the past decade, significant progress has been made, as evidenced by the availability of databases with multi-omics data and software to integrate genome sequences and annotation enabling us to broaden ecological and applied research (e.g., PhycoCosm, which integrates genome sequences and annotation for >100 algal genomes, [146]). Aided by AI, it is expected that databases and software will continue developing and improving. As such, the field of microalgal proteomics is on the edge of a huge acceleration of data generation and new opportunities. For instance, while combining omics techniques (multi-omics) may prove critical to building a detailed picture of molecular signatures, their interactions and the phenotypes [16,18,22,24], multi-omics data integration is currently hindered by the heterogeneous nature of data across multi-omics datasets and the natural noise in biological data. AI technology could play a critical role in addressing these challenges [24].

Proteomics has been instrumental improving our understanding of microalgal biology. The continuous development of sample preparation techniques (e.g., extraction, proteases) and high-throughput techniques via AI-based pipelines and software will provide a suitable platform to decipher complex metabolic processes in microalgae. Importantly, many chemicals can be produced by microalgae; therefore, future studies should continue focusing on conditions other than the ones used to trigger lipids production. In addition, the study of PPIs can significantly improve our understanding of protein function, interaction and cellular processes, as shown for the *C. reinhardtii* fatty acid metabolism [84]. Further PPI analyses of key microalgal proteins could, therefore, lead to significant advance for strain (or protein) selection and modifications.

Advances in genetic engineering (e.g., CRISPR/Cas9) have facilitated the creation and analysis of mutants, offering deep insights into vital functions, such as those of global regulators [121,147]. The field of synthetic biology and the production of protein of interest by heterologous expression should therefore continue advancing biological knowledge while also providing the opportunity for new biotechnological applications with the production of proteins of biotechnological interest.

## 8. Conclusions

In summarizing the recent advances in microalgal proteomics, this review underscores the crucial role of these technologies in understanding both fundamental biological processes and their biotechnological applications. Yet the importance of rigorously maintaining key steps from sample preparation to data analysis for successful proteomics analyses cannot be overstated. Recent years have seen substantial efforts to develop and streamline proteomics frameworks, making this field increasingly accessible to non-experts. This evolution has been complemented by the emergence of companies specializing in affordable data analysis, curation and visualization. With the ongoing integration of AI, these processes are set to become even more efficient and reliable. The insights gained from functional enrichment analyses and protein–protein interactions have been instrumental in addressing complex biological questions and will certainly guide future research directions (e.g., strain improvement). As we transition into the post-omics era, the increasing availability of advanced software and databases promises a broad spectrum of discoveries in both fundamental and applied microalgal research. This journey into the future of microalgal proteomics is not just about enhancing our understanding but also about harnessing the potential of microalgae in innovative and sustainable ways. Notably, microalgae can produce a wide array of chemicals (e.g., nutraceuticals, pharmaceuticals); therefore, the field would benefit from more proteomics studies focusing on conditions triggering the production of these chemicals.

## Figures and Tables

**Figure 1 proteomes-12-00013-f001:**
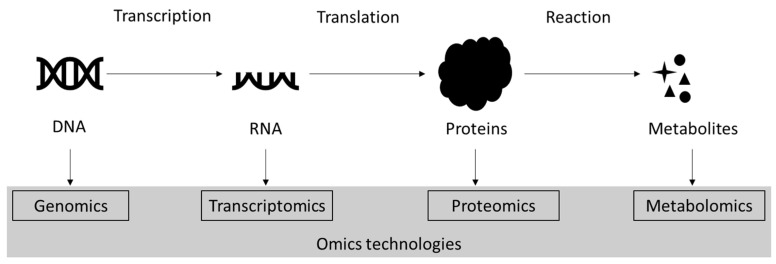
The four main omics. While not represented on the figure, the study of the interactions between and among proteins and other molecules within a cell is referred as interactomics.

**Figure 2 proteomes-12-00013-f002:**
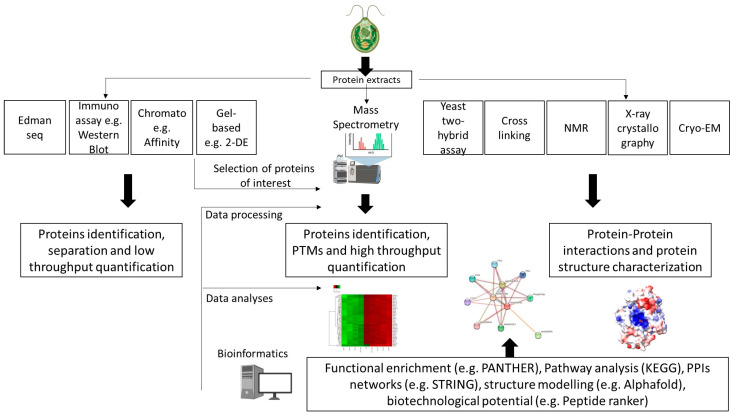
Proteomics techniques and tools commonly used for microalgae research (seq: sequencing; Immuno: immunodetection; Chromato: chromatography; PTMs: post-translational modifications).

**Figure 3 proteomes-12-00013-f003:**
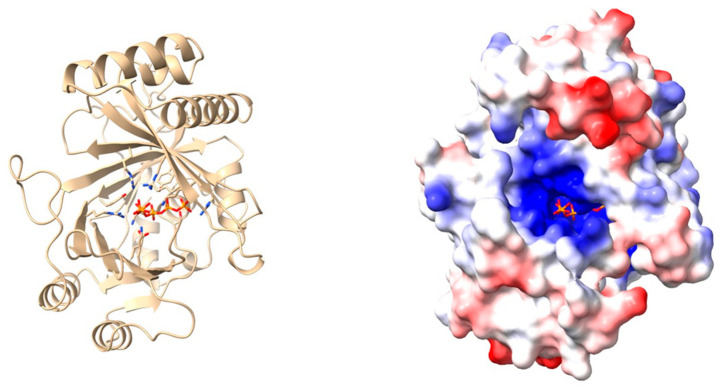
*Gonium pectorale* VTC4 catalytic core domain, shown in complex with a polyP5 molecule (left, cartoon representation, right, surface colors mapped to the electrostatic potential (−10: red to 10: blue KbT/e) at pH 7.0 of the apo form). The models were adapted from Cliff et al. [41] under a Creative Commons license (CC BY-NC-ND 4.0 DEED).

**Table 1 proteomes-12-00013-t001:** Overview of the methodologies available to study proteins/proteome with references to their use in *C. reinhardtii*.

	Method	Description	Example(s)
Conventional	Chromatography	Chromatography-based techniques are used for protein separation and purification. In brief, proteins can be separated/purified based on charges (IEC: ion-exchange chromatography), size (SEC: size-exclusion chromatography) or (bio)chemical affinity with a matrix (AC: affinity chromatography).	[63,64,65]
	Immuno assay/blotting	The enzyme-linked immunosorbent assay (ELISA) detects the presence of a protein by measuring the enzymatic activity of an enzyme-labeled antigen or antibody binding to an immobilized target protein (or antibody). ELISA is widely used for diagnostics. Western blotting allows the separation of proteins based on molecular weight through gel electrophoresis and identification of protein from binding of a labeled antibody to its target antigen (i.e., protein) on a membrane.	[66,67]
	Edman sequencing	The Edman method (or Edman degradation) determines the amino acid sequence in peptides/proteins by sequentially identifying and cleaving amino acids from the N-terminal side of a peptide/protein.	[68]
Advanced	Gel-based	The conventional polyacrylamide gel-based method used for protein separation and identification based on proteins’ mass (SDS-PAGE: sodium dodecyl sulfate–polyacrylamide gel electrophoresis) has evolved into more advanced 2D methods based on both mass and charge separation (2D-PAGE: two-dimensional polyacrylamide gel electrophoresis) or using labels with a fluorescent dye (2D-DIGE: two-dimensional differential gel electrophoresis). 2D-PAGE and 2D-DIGE can resolve and investigate the abundance of several thousand proteins in a single sample.	[59,62,69]
	Mass spectrometry	MS is one of the most used analytical techniques to identify and, coupled with (ultra)-high performance chromatography, quantitatively measure protein levels. MS of peptides/proteins ionized via matrix-assisted laser desorption ionization (MALDI), surface-enhanced laser desorption/ionization (SELDI) or, more classically, electrospray ionization (ESI), allow the determination, through deconvolution of the mass spectra obtained, of their molecular mass. In the context of proteomics and peptide analysis, multi-stage tandem mass spectrometry (MSn) is also a powerful technique for obtaining detailed information on the structure and sequence of peptides and proteins, particularly with regard to the localization of post-translational modifications (PTMs). Label-free or labeled approaches can be used for quantification (ICAT: isotope-coded affinity tag; iTRAQ: isobaric tagging for relative and absolute quantification; SILAC: stable isotope labeling by/with amino acids in cell culture; QconCAT: quantification concatemer; TMT: Tandem Mass Tag). MS is often combined with separations and fractionation techniques to identify target proteins or subproteomes. Sample fractionation and enrichment are also important when identifying PTMs (phosphorylation, oxidation, nitrosylation, glycosylation, methylation, etc.).	ICAT: [46] iTRAQ and label-free: [70] iTRAQ: [71] QconCAT: [72] SILAC: [50] TMT: [40] PTMs: [46,47,51,53,73]
	Nuclear Magnetic Resonance	Protein structural determination in solution or solid phase by measuring chemical shifts. NMR structural determination involved several steps, each requiring significant expertise/techniques.	[74]
	X-ray crystallography	Three-dimensional protein structures are determined by exposing highly purified crystallized protein samples to X-rays and measuring diffraction patterns.	[75]
	Cryogenic electron microscopy	Microscopy techniques used to determine 3D structure of proteins or protein complexes via flash freezing and electron bombardment of samples in solution.	[55]
	In silico modelling	Generation and study of protein 3D models using homology modelling or deep-learning-based predictions.	[41]

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
