# Peer review of "Key Proteomics Tools for Fundamental and Applied Microalgal Research"

_proteomes, 2024, doi:10.3390/proteomes12020013_

Round 1

Reviewer 1 Report

Comments and Suggestions for Authors

1.    The brief introduction and emphasis of proteomics, including from the main quantitative and qualitative approaches and their applications in microalgal studies should be summarized in the abstract section, instead of just omics.

2.    In the introduction section, the author should include some comparisons of proteomics and other omics and the specialty of the proteomics, thus the readers can expect the following review.

3.    A summarized graph describing the whole proteomic technologies used in microalgal studies can help readers understand its current status.

4.    Not just Chlamydomonas reinhardtii, the author should include those most commonly studies microalgal organisms, including Haematococcus, Dunaliella, Chlorella, and other economic and ecological species.

Comments on the Quality of English Language

1.    The brief introduction and emphasis of proteomics, including from the main quantitative and qualitative approaches and their applications in microalgal studies should be summarized in the abstract section, instead of just omics.

2.    In the introduction section, the author should include some comparisons of proteomics and other omics and the specialty of the proteomics, thus the readers can expect the following review.

3.    A summarized graph describing the whole proteomic technologies used in microalgal studies can help readers understand its current status.

4.    Not just Chlamydomonas reinhardtii, the author should include those most commonly studies microalgal organisms, including Haematococcus, Dunaliella, Chlorella, and other economic and ecological species.

Author Response

We thank Reviewer 1 for his/her review.

  1. The brief introduction and emphasis of proteomics, including from the main quantitative and qualitative approaches and their applications in microalgal studies should be summarized in the abstract section, instead of just omics.

The abstract has been modified. We have now included the following Li 14 - 17:

” In particular, where studies focused on a single or few proteins decades ago, it is now possible to study the whole proteome of a microalgae. The development of mass spectrometry-based methods has provided this leap forward with the high-throughput identification and quantification of proteins.”

  1. In the introduction section, the author should include some comparisons of proteomics and other omics and the specialty of the proteomics, thus the readers can expect the following review.

The introduction has been modified accordingly (Li 37 – 38, Li 52 – 60) and Figure 1 has been added to improve clarity.

  1. A summarized graph describing the whole proteomic technologies used in microalgal studies can help readers understand its current status.

We have included Figure 2 (Li 178) that present the technologies used in microalgae proteomic research.

  1. Not just Chlamydomonas reinhardtii, the author should include those most commonly studies microalgal organisms, including Haematococcus, Dunaliella, Chlorella, and other economic and ecological species.

We focused on C. reinhardtii because it is a model organism, but we are now discussing and citing studies focusing on species commercially produced or with commercial potential see Li 113-114, Li 265-266, Li 283, Li 380, Li 457 - 461, Li 470-471.

Reviewer 2 Report

Comments and Suggestions for Authors

I carefully assessed this work, and I am honestly sorry to inform you that the manuscript is not suitable for the requirements of this journal. The introduction should be improved and more concretely written.

The whole manuscript lacks deeper analysis, and content organization, and seems simple in content quality. In several places, authors presented only fundamental concepts and information and a deeper analysis of regulation mechanisms. Discussion and conclusion must be carefully improved and future perspectives added to the research. 

The quality and novelty of the manuscript are relatively low. I would have expected a more critical discussion. The paper is very pragmatic.

Comments on the Quality of English Language

I carefully assessed this work, and I am honestly sorry to inform you that the manuscript is not suitable for the requirements of this journal. The introduction should be improved and more concretely written.

The whole manuscript lacks deeper analysis and content organization and seems simple in content quality. In several places, authors presented only fundamental concepts and information and a deeper analysis of regulation mechanisms. Discussion and conclusion must be carefully improved, and future perspectives must be added to the research. 

The quality and novelty of the manuscript are relatively low. I would have expected a more critical discussion. The paper is very pragmatic.

Author Response

I carefully assessed this work, and I am honestly sorry to inform you that the manuscript is not suitable for the requirements of this journal. The introduction should be improved and more concretely written.

The whole manuscript lacks deeper analysis, and content organization, and seems simple in content quality. In several places, authors presented only fundamental concepts and information and a deeper analysis of regulation mechanisms. Discussion and conclusion must be carefully improved and future perspectives added to the research. 

The quality and novelty of the manuscript are relatively low. I would have expected a more critical discussion. The paper is very pragmatic.

We thank reviewer 2 for his/her time and honesty.

The aim of our review was to give an overview of the proteomics technologies and their applications for microalgal research to provide readers with a better idea of what is available (Li 61 – 62). We did not extensively detail some parts which have been reviewed by others, we referred to this key literature instead (e.g. Li 283). However, we stressed and discussed the importance of protein extraction (Li 257 – 283), critical for microalgal research, as well as the use and potential of Artificial intelligence in the field for large-scale dataset processing, analyses (Li 307 – 395) and to develop/evaluate new biotechnological application for proteins (Li 398 – 409; Li 492 – 551).

Nevertheless, following Reviewer 2` recommendations we have now improved the introduction, discussion and conclusions (Li 14-17; Li 37; Li 52 – 60, Section 3.1 Quantitative high-throughput proteomics: Mass Spectrometry and 3.2 Protein-protein interactions techniques Li 182-235, and see Li 285-306 and Li 327-332 for further detail about MS process and sample preparation). The overall organization of the manuscript was also improved (we swapped Section 4 and 5 and see small changes throughout the manuscript). Finally, we included a future prospect section (Li 557 – 587) and we have re-written the conclusions.

Reviewer 3 Report

Comments and Suggestions for Authors

Overall, the manuscript is well-written. No major concerns were found in the manuscript. My comments are given below.

·        It’s an interesting manuscript which provides the comprehensive review about the proteomics in the microalgae field.

·        One or two images would be helpful to summarize the overall concept.

·        Section 5.1. Sample preparation looks fine. I have found that authors have given little information about the mass spectrometry process in the section 5.2. It would be more helpful to the readers if the authors provide more information about the MS sample preparations like in-gel digestion or in-solution digestion followed by protein sample preparation.

·        Bioinformatics part should be explained more. More free online tools like STRING, GO, etc. must be included. Or if any microalgae specific databases/online prediction tools must be included.

·        Few more applications like protein-protein interaction techniques should be included in the manuscript.

Comments on the Quality of English Language

English looks fine to me in general sense. Few typos and grammatical errors must be corrected. 

Author Response

Overall, the manuscript is well-written. No major concerns were found in the manuscript. My comments are given below.

  • It’s an interesting manuscript which provides the comprehensive review about the proteomics in the microalgae field.

We thank Reviewer 3 for his/her review and for praising our manuscript.

  • One or two images would be helpful to summarize the overall concept.

Two figures have been included to 1. Summarize the overall concept of proteomics among the other omics technologies. And 2. Present common proteomics techniques applied in microalgal research.

  • Section 5.1. Sample preparation looks fine. I have found that authors have given little information about the mass spectrometry process in the section 5.2. It would be more helpful to the readers if the authors provide more information about the MS sample preparations like in-gel digestion or in-solution digestion followed by protein sample preparation.

We have provided more information about MS process and sample preparation (Li 182-207, Li 285-306 and Li 327-332)

  • Bioinformatics part should be explained more. More free online tools like STRING, GO, etc. must be included. Or if any microalgae specific databases/online prediction tools must be included.

We have re-organized the bioinformatics section for clarity, and we have included the databases/tools recommended by Reviewer 3 (Li 365 – 369, Li 380 – 382, Li 383 – 389, Li 563 – 564).

  • Few more applications like protein-protein interaction techniques should be included in the manuscript.

We have included a section discussing protein-protein interaction techniques and their potential for microalgal research (Li 209-235).

Round 2

Reviewer 3 Report

Comments and Suggestions for Authors

The authors responded well to the queries. I recommend the journal editor to publish this manuscript in the current format. One small suggestion to the authors. The term "Interactomics" in the legend of Figure 1 needs to be revised as it implies the study of interactions between molecules. However, the message conveyed in the figure does not align with this definition. I feel “Omics technologies” would be a better term. However, I leave it to the authors either to correct it or keep it as such.  

Author Response

We thank Reviewer 3 for his/her review and recommendation about our manuscript.

We agree with Reviewer 3 and we have made the suggested change in the Figure. For clarity, we have also modified the Figure caption to:

Figure 1. The four main omics. While not represented on the figure, the study of the interactions between and among proteins and other molecules within a cell is referred as interactomics.